# Lithographically-Fabricated HA-Incorporated PCL Nanopatterned Patch for Tissue Engineering

**Kyoung-Je Jang [1]**, **Sujin Kim [2]**, **Sangbae Park [1]**, **Woochan Kim [2]**, **Yonghyun Gwon [2]**,
**Sunho Park [2]**, **Ki-Taek Lim [3]**, **Hoon Seonwoo [4],\*** and **Jangho Kim [2],\***

[1] Department of Biosystems & Biomaterials Science and Engineering, Seoul National University,
Seoul 151-742, Korea; trudwp@gmail.com (K.-J.J.); sb92park@snu.ac.kr (S.P.)

[2] Department of Rural and Biosystems Engineering, Chonnam National University, Gwangju 61186, Korea;
kimsujin4172@gmail.com (S.K.); rladncks92@gmail.com (W.K.); gyhjhj0716@gmail.com (Y.G.);
preference9330@gmail.com (S.P.)

[3] Department of Biosystems Engineering, College of Agricultural and Life Sciences, Kangwon National
University, Chuncheon 24341, Korea; ktlim@kangwon.ac.kr

[4] Department of Industrial Mechanical Engineering, College of Life Science and Natural Resources,
Sunchon National University, Sunchon 57922, Korea

\* Correspondence: uhun906@gmail.com (H.S.); rain2000@jnu.ac.kr (J.K.)

**Abstract:** Inspired by the aligned extracellular matrix and bioceramics in human bone tissue, we investigated the relative contributions of nanotopography and equine bone powders (EBPs) with human dental pulp stem cells (DPSCs) to the osteogenesis. Both nanotopography and EBPs independently promoted the osteogenesis of DPSCs, osteogenesis was further promoted by the two factors in combination, indicating the importance of synergistic design factor of guided bone regeneration (GBR) membrane. The osteogenesis of DPSCs was affected by the polycaprolactone-based nanotopography of parallel nanogrooves as well as EBPs coating. Interestingly, both nanopattern and EBPs affected the DPSCs morphologies; nanopattern led to cell elongation and EBPs led to cell spreading and clustering. Analysis of the DPSCs-substrate interaction, DPSCs-EBPs interaction suggests that the combined environment of both factors play a crucial role in mediating osteogenic phenotype. This simple method to achieve a suitable environment for osteogenesis via nanotopography and EBPs coating modulation may be regarded as a promising technique for GBR/GTR membranes, which widely used dental and maxillofacial surgery applications.

**Keywords:** nanogrooves; PCL patches; equine bone powder; dental pulp stem cells

## 1. Introduction

Biomaterials-based scaffolds have garnered interest in medical fields, such as cell transplantation, tissue engineering, regenerative medicine, and drug delivery [1,2]. In particular, the use of biomaterials to produce in vivo like micro- and nano-environments to control cellular behaviors and functions remains a challenging task. Nanotopographically defined scaffolds (e.g., highly aligned nanomatrix) have been proposed based on previous studies indicating that living cells are highly sensitive to the local architecture of the complex and well-defined structures of the extracellular matrix (ECM) [3,4]. It means that the architecture of synthetic extracellular matrices (ECMs) inspired by target tissue for programmed stem cell behavior and response, such as proliferation and differentiation could be essential design factors of tissue-engineered scaffolds. Despite various efforts in designing physicochemical structures of native tissues, such as bone [5], tendon [6,7], ligament [7], tooth, and periodontal tissue, optimal scaffolds remain to be developed, given the complex environment of in vivo tissues [8].

The composition of in vivo-like synthetic nanostructured matrices to promote stem cell function and behavior, such as attachment, proliferation and differentiation, has been regarded as a topic of interest in the field of tissue regeneration; it is important to control the surface topography at a nanoscale resolution [9,10]. Currently, various technologies for micro-, nanofabrication such as electron beam- or photolithography [11], self-assembling systems [12], micro-contact printing [13], particle synthesis [14], replica casting or modeling [15], chemical etching [16], sandblasting [17], and electrospinning [18] have advantages and limitations respectively. For example, electrospun fiber mesh has the advantage of being easy to obtain non-woven, anisotropic aligned mesh consisting of large amounts with nanoscale fibers using various biocompatible polymers. However, they have some drawbacks as a substrate for cell cultures such as difficulties to penetrate cells between mesh and to have a consistent thickness of fibers in mesh, as well as having precisely defined nanopatterns. Compared to the electrospinning, it has been known that the soft lithographic techniques are more precise, achieving 2D matrices, and can circumvent low yield of electrospinning technique.

The typical forms of nanotopographical pattern—nanogroove, nanopits, and nanowire—could be generated by using soft lithography techniques. Especially, it was found that nanogrooves patterned substrate could enhance initial cell extension, direct cell elongation (orientation), controlling cell migration along nanogrooves, and promote cell differentiation. Abagnale et al observed that in the presence of differentiation media, nanogrooves (650 nm) enhanced mesenchymal stem cells (MSCs) differentiation into both osteogenic and adipogenic lineages. In addition, Kim et al observed the NIH/3T3 cell migration on the anisotropic nanotopographic pattern arrays with variable local sizes (300-800 nm), and the cell migration was most rapidly on the dense nanopattern (300 nm) [19]. These findings have supported the importance of nanotopographical features for stem cell fate and biomedical applications.

Calcium phosphate (CaP) is a major inorganic component of animal bones and teeth. Hydroxyapatite (HAp), biphasic calcium phosphate (BCP), and β-tricalcium phosphate (β-TCP) have been known as the most representative materials of CaP, which is known as a material that is well integrated with adjacent bone tissue that can promote bone regeneration [20,21]. In particular, CaP minerals extracted from animal bone tissues are known as xenografts, which have a similar crystal structure to hydroxyapatite and biocompatible due to its trace elements and similarity of chemical composition to human bone [22,23]. Therefore, the large-sized (0.2–1 mm) CaP mineral extracted from xenogeneic sources has been widely used a as clinical orthopedic and as dental implants to fill defected bone tissue regions. The nanosized CaP mineral from exogenous could be used as a coating on polymeric or metallic scaffolds to provide an enhanced cellular response [24]. It has been reported that calcium minerals would be effective against the periodontitis and gingivitis when applied with vitamin D [25,26].

Polycaprolactone (PCL) is a promising biomaterial for the production of the Food and Drug Administration (FDA)-approved scaffolds that can be used in tissue engineering and regenerative medicine, especially for GBR/GTR (guided bone regeneration/guided tissue regeneration) membrane such as a membrane for periodontal tissue [27]. However, the clinical usage of PCL-based scaffolds is still limited due to their poor properties, such as low bioactivity, strong hydrophobicity, and weak inflammatory reaction [28,29], which hindered the cellular response [30]. In particular, periodontitis and gingivitis could promote the breakdown of periodontal tissue and induce the loss of periodontal tissue [31]. In particular, it has been reported that endothelin (ET), and ET-1 are factored that influence the formation of collagen, a representative ECM protein [32], showing that the high blood levels of ET-1 affected host defense substrates, negatively affecting dental-related diseases [33].

Here, we hypothesize that (i) the nanotopographical and equine bone powders (EBPs) cues would enhance cellular behaviors (e.g., osteogenesis) and (ii) the cellular behaviors could be further enhanced by two factors in combination, which can be an efficient strategy for design and fabrication of biomedical patches. To this end, we developed lithographically defined PCL-based nanogroove patterned patches with EBPs, using capillary force lithography in combination with a dip-coating

method without structural integrity of nanogroove pattern. We examined the role of mechanically patterned substrate and EBPs using human-derived dental pulp stem cells (DPSCs) by analyzing morphological features and osteogenesis.

## 2. Materials and Methods

### 2.1. Preparation of Nanogrooved PCL Patches

The poly (urethane acrylate) (PUA, Chansung Sheet., Korea) mother mold (800 nm ridge and groove width) and the PDMS (Polydimethylsiloxane) replication mold were prepared as previously reported [10]. The thin polycaprolactone (PCL, Mw: 80,000; Sigma-Aldrich, USA) patch was fabricated via spin-coating (rotator speed: 3500 rpm, duration time: 120 secs, acceleration: 5 secs) using PCL solution (18 wt% in dichloromethane). The nanopatterned PDMS mold was placed on the pre-melted PCL layer and embossed onto the melted PCL layer by applying pressure while heating at 80 °C for 2 mins. Then, the assembly which is composed of the PCL layer and PDMS mold was cooled at 25 °C for 30 mins, and the PDMS mold was peeled off from the PCL layer (NG-PCL). In addition, the flat PCL patches were prepared using the flat PDMS molds (F-PCL).

### 2.2. Fabrication of EBP-Coated Nanogrooved PCL Patches

The femur of a 24-month-old equine was prepared [9]. The thermally treated equine bone (900 °C, 4 h) was ground using a high energy ball mill (Planetary Mono Mill, Pulverisette-6, Fritsch, Germany). Ethanol (EtOH) and dichloromethane (DCM) were mixed at a ratio of 95:5 (v:v). EBPs were dispersed in 2 wt% EtOH/DCM solution with sonication for 5 mins. The NG-PCL was soaked in the EtOH/DCM solution for 20 s and 1, 5, and 10 mins. The NG-PCLs were then removed from the EBPs solution and dried at room temperature for 2 hours. The dried NG-PCLs were washed twice with deionized water, and named NG-PCL-C. The same coating protocol was conducted to the F-PCL. Then, the equine bone powder-coated F-PCL was named as a F-PCL-C.

### 2.3. The Owen-Wendt Method

The Owen-Wendt method is used to determine the surface free energy (SFE) of solid. Based on Bethelot's hypothesis, this method is based on contact angle measurements and is the most widely used method for SFF calculations. The following equation is given to determine the SFE:

$$\gamma_S = \gamma_S^d + \gamma_S^p,$$

where $\gamma_S$ is the surface free energy (SFE), $\gamma_S^d$ is the dispersion component of SFE and $\gamma_S^p$ is the polar component of SFE.

To determine the SFE, two measurement liquids were used (deionized water and diiodomethane) to observe the polar and dispersion component of the surfaces. The contact angles of each specimen by two liquids were measured respectively. The SFE was calculated using the following equation:

$$\left(\gamma_S^d\right)^{0.5} = \frac{\gamma_d(\cos\Theta_d + 1) - \sqrt{\left(\frac{\gamma_d^p}{\gamma_w^p}\right)}\gamma_w(\cos\Theta_w + 1)}{2\left(\sqrt{\gamma_d^d} - \sqrt{\gamma_d^p\left(\frac{\gamma_w^p}{\gamma_w^p}\right)}\right)}.$$

$$\left(\gamma_S^p\right)^{0.5} = \frac{\gamma_w(\cos\Theta_w + 1) - 2\sqrt{\gamma_s^d\gamma_w^d}}{2\sqrt{\gamma_w^p}}.$$

In order to calculate the SFE of each specimen, the contact angles $\Theta$ with the surface of distilled water and diiodomethane ($CH_2I_2$, Sigma Aldrich, USA, St. Louis) were measured [34].

## 2.4. Observation of Cellular Behaviors on EBP-Coated Nanogrooved PCL Patches

The human-derived dental pulp stem cells (DPSCs) were isolated from the dental tissue of 13 adult patients for treatment purposes (Intellectual Biointerface Engineering Center, Dental Research Institute, College of Dentistry, Seoul National University). The DPSCs were grown in alpha-MEM containing 10% FBS and 1% penicillin–streptomycin (Gibco, Milan, Italy) in the 96-well plate with each sample respectively. After 12 h and 7 days of culture, the DPSCs viability on each sample patches were evaluated using a WST-1 assay kit (Daeillab service, Seoul, Korea). To remove the unattached cells in media, the remained culture media were removed and washed with phosphate-buffered saline (PBS, Gibco, Milan, Italy) before the cell viability measurement. The culture medium and WST-1 solution were mixed at a ratio of 1:10 (WST-1: culture media). The 100 µl of the mixed solution was put into each wellplate. The WST-1 treated samples were incubated at 37 °C, $CO_2$ 5%, and 95% RH condition for 1 hour. The supernatant 50 µl of the reaction media was transferred to a new 96-well plate to observe the transmittance at 450 nm. Each group was divided by the absorbance value of F-PCL for normalization.

After 7 days of culture, cell cytoskeletons were observed under a microscope (Eclipse, Nikon, Japan) through immunofluorescence for β-actin. The morphology of DPSCs on the patches was observed using FE-SEM (Field Emission Scanning Electron Microscope, Supra, Carl Zeiss, Germany), and the specimens were prepared using vapor fixation method [35].

For osteogenic differentiation studies, the cells were cultured in the osteogenic induction medium (OM) containing 0.1 µM dexamethasone, 10 mM sodium β-glycerophosphate, and 0.05 mM ascorbic acid-2-phosphate (Sigma-Aldrich, St. Louis, MO, USA). For alkaline phosphatase activity assessment, by using 100 µL of Triton-X the total protein was extracted from DPSCs cultured on each specimen at 7 days during the period of study. For sedimentation of cell debris, the extracted solution was centrifuged at 10,000 rpm at 4 °C for 15 mins. After centrifugation, the supernatant was collected and ALP activity was evaluated with an ALP assay kit (SensoLyte pNPP Alkaline Phosphatase assay kit, AnaSpec, Fremont, CA).

## 2.5. Analyzing the Cellular Morphology

The morphology of DPSCs on each sample was analyzed using FE-SEM and fluorescent images. The Image J software was used for measuring cellular length and width. The cellular length is the longest line that passes by the nucleus. The cellular width is the shortest line that passes by the nucleus. The cell elongation factor was calculated using the equation as follows:

$$\text{Cell elongation factor} = \frac{\text{Cellular length}}{\text{Cellular width}}.$$

The multi-cell outliner that is the ImageJ plugin was used to quantify the cell area. The cellular edge was detected by this software and the cellular area was calculated using the number of pixels in the image.

## 2.6. Statistical Analyses

The data were expressed as the means and standard deviations (SDs). The statistical analyses were performed with R software (RStudio, Inc, Boston, Ma) and analyses of variance combined with one-way ANOVAs. Differences with p values < 0.05 were considered statistically significant.

## 3. Results

### 3.1. Preparation of NG-PCL-C

The lithographically fabricated NG-PCL and F-PCL were coated with the equine bone powders with submicron-size (EBPs). The detailed characteristics of EBPs are shown in Figure S1. The cumulative diameters (d0.1, d0.5, d0.9) of EBPs are 0.779 ± 0.099, 4.13 ± 1.71 and 19.40 ± 0.2.47 μm, respectively. Figure 1a shows a schematic diagram of the nanogrooved PCL patch (NG-PCL) fabrication process and EBPs coated NG-PCL with dip-coating method. Figures S2 and S3 show the optimization of coating conditions. In our previous study, we observed the surface morphologies of native ECM of the human body, which was well aligned of the bunch of the ECM fibers [9]. And the lithography technique was used for manipulating the naonogroove/ridge of NG-PCL. As shown in Figure 1b, the fabricated NG-PCL was confirmed to contain well-defined nonogrooves/ridges regularly. In addition, the EBPs were well coated on the NG-PCL (Figure 1b) without damaging the nanogroove/ridge structure of NG-PCL (e.g., designed nanostructures) during the dip-coating process, and were well dispersed on the surfaces of NG-PCL. Even though the surface of the patch was covered with EBPs, the nanopatterns were still observed (Figure S4).

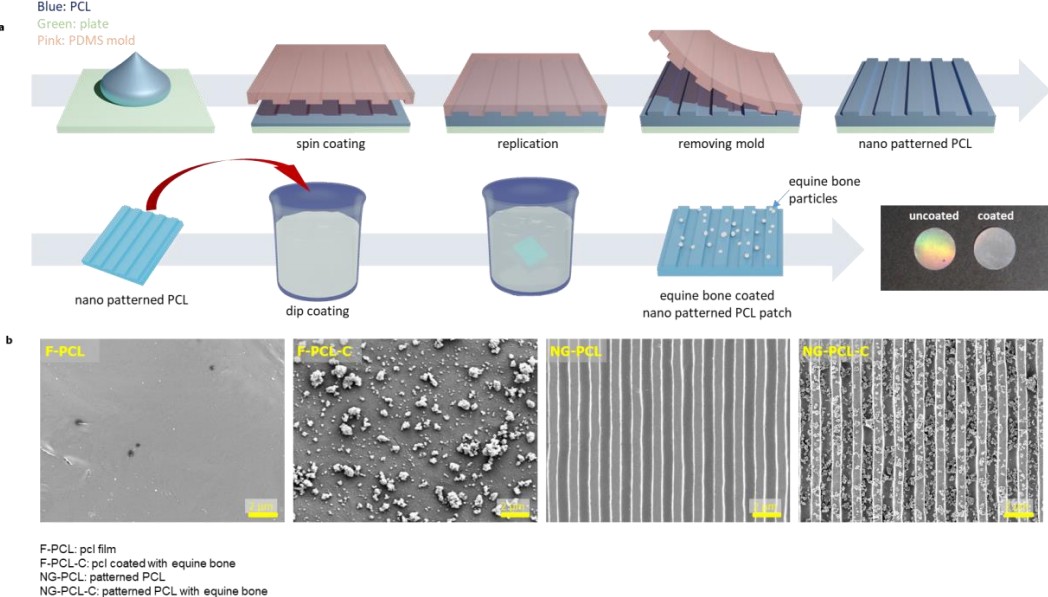

**Figure 1.** Preparation of PCL flat/nano-patterned patches coated with equine bone powders. (**a**) The schematic diagram of PCL nano-patterned patches and equine bone powder-coated PCL nano-patterned patches. (**b**) Surface observation of PCL flat and nano-patterned patches with or without equine bone powders.

The water contact angle was measured to prove the effectiveness of the EBP coating for enhancing hydrophilicity (Figure 2). Interestingly, the water droplet forms an ellipse shape along with the direction of the nanogroove patterns. In addition, the water contact angle at the frontal and lateral direction of the droplet showed a different value (Figure 2a,b). The water contact angle of the nanopattern patch was measured in the major axis direction of the elliptical droplet and in perpendicular directions. On the other hand, the water droplet shape was an almost circular shape on the F-PCL and F-PCL-C (Figure 2a). Furthermore, it is noted that the water droplet elongation factor of NG-PCL-C was smaller than the value of NG-PCL. However, the elongation factor of NG-PCL-C was 1.20 and it was higher than those of F-PCL and F-PCL-C. This means that the droplets in NG-PCL-C forms an elliptical shape. In addition, the water contact angles of the NG-PCL and NG-PCL-C were higher than those on the F-PCL and F-PCL-C respectively, due to the nanogrooved surface topography. The EBPs coating enhanced the

hydrophilicity; thus it led to decreasing the water contact angles of each sample. The elliptical water droplet on the NG-PCL and NG-PCL-C had different water contact angles along with their direction. Interestingly, the angles at the minor axis direction did not show a significant difference with the value of F-PCL and F-PCL-C. Furthermore, it was found that the contact angles at the major axis direction were more decreased than that of the minor axis.

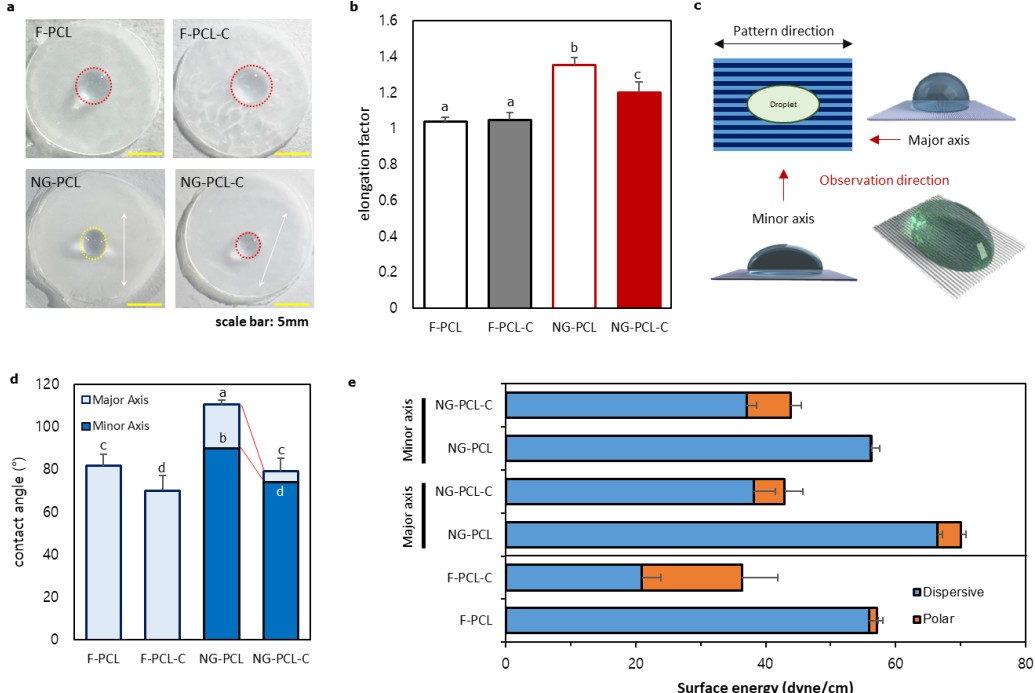

**Figure 2.** Measurement of wettability of PCL flat and nano-patterned patches with or without equine bone powders. (**a**) Morphology of water droplet, (**b**) elongation factor of the water droplets (n=7) (the ratio of long/short axis length of the water droplet), (**c**) schematic of observation direction of water contact angle. (**d**) The water contact angle on the flat patches (F-PCL and F-PCL-C) was constant regardless of the observation direction, and in the nanogroove pattern (NG-PCL, NG-PCL-C), there was a difference in the observation direction; the water contact angle in both cases was measured. (n = 7) (**e**) The results of calculated surface energy based on the Owen–Wendt method. Orange bars mean the surface energy by polar component and blue bars mean the surface energy by non-polar component. It was observed that the total surface energy was decreased by equine bone powder coating, and the surface energy by polar components was increased.

### 3.2. Cellular Behaviors on the DPSCs in the Equine Bone Coated Nanopatterned Patch

To evaluate the initial cellular behaviors, we cultured DPSCs on the substrates for 12 h (Figure 3). The DPSCs on the patterned substrates were aligned along the nanogroove direction, whereas DPSCs attached to flat substrates were not arranged in a particular direction (Figure 3a). EBP coating affected the cell morphology of DPSCs in nanopattern and flat patches. DPSCs in the flat patch had smaller cell adhesion areas than other groups, whereas DPSCs in the F-PCL-C had a larger cell area than flat patches and DPSCs formed clusters around the coated EBP powder. DPSCs attached to the surface formed with the nanopatterns were observed to be elongated along with the nanogroove direction. The samples which treated with EBP coating, and more cells were attached to stray from the direction of the pattern (Figure 3a). A 12-hour cell viability evaluation experiment confirmed that all four groups had similar values (Figure 3b).

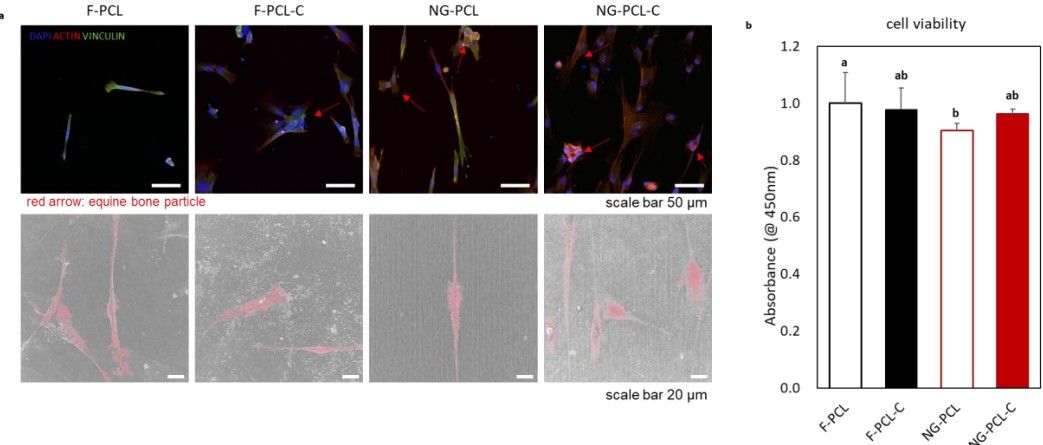

**Figure 3.** Observation of cellular behavior of DPSCs cultured for 12 hours. (**a**) Representative fluorescence images of DPSCs on the substrate and red arrows indicate the EBPs (upper row), FE-SEM images of DPSCs on the substrate and pink color indicate DPSCs (lower row). (**b**) The result of DPSCs cell viability at 12 hours of cell culture (n = 5). Bars with the same letter are not significantly different according to Duncan's multiple range test at 5% probability.

To observe the time-dependent morphologies of DPSCs on the substrate, we cultured the DPSCs on the nanopatterned matrix with EBPs for 1 and 7 days (Figure 4) and the images were obtained through FE-SEM and CLSM (Confocal laser scanning microscopy). The DPSCs morphology on NG-PCL-C at day 7 was especially well-oriented along with the nanogroove pattern compared to day 1. In addition, we can observe the expanded cellular shape on the EBPs coated substrate. Although DPSCs on F-PCL and F-PCL-C densely populated, we observed that the cellular morphologies on the substrate were randomly aligned at day 7.

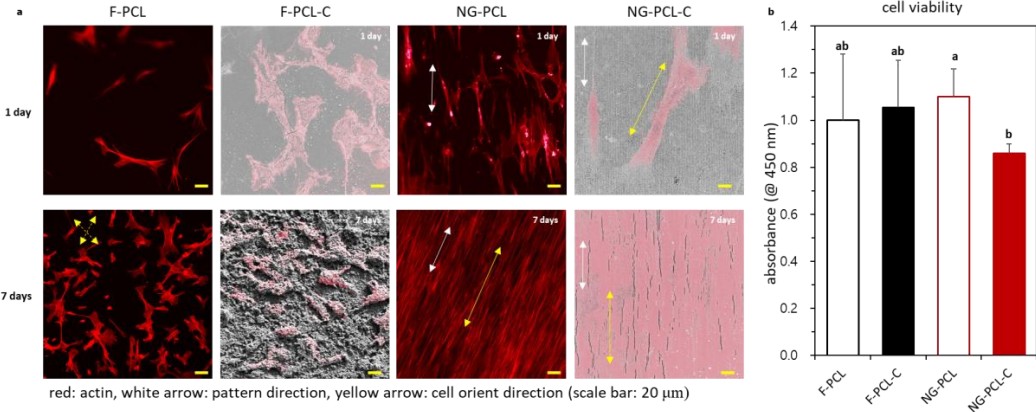

**Figure 4.** DPSCs viability for 7 days of cell culture. (**a**) Observation of the DPSCs morphologies 1 and 7 days of cell culture. (**b**) The results of DPSCs cell viability at 7 days for the cell culture (n =5). Bars with the same letter are not significantly different according to Duncan's multiple range test at 5% probability.

To evaluate the combined effect of nanopatterns and EBPs, we cultured DPSCs using the osteogenic conditioned media for 2 weeks. The results of the ALP activity (Figure 5a), the initial osteogenic differentiation marker, showed significant differences in NG-PCL-C compared to F-PCL-C group. As shown in Figure 5c, differences in calcium mineral deposition were observed by Alizarin Red S (ARS) staining. The DPSCs cultured on F-PCL-C and NG-PCL-C showed more accumulated calcium deposition than those on the uncoated substrate (i.e., without EBPs) (Figure 5b). It is also found that the DPSCs cultured on the NG-PCL and NG-PCL-C showed more calcium deposition than those on the F-PCL and F-PCL-C respectively (Figure 5b).

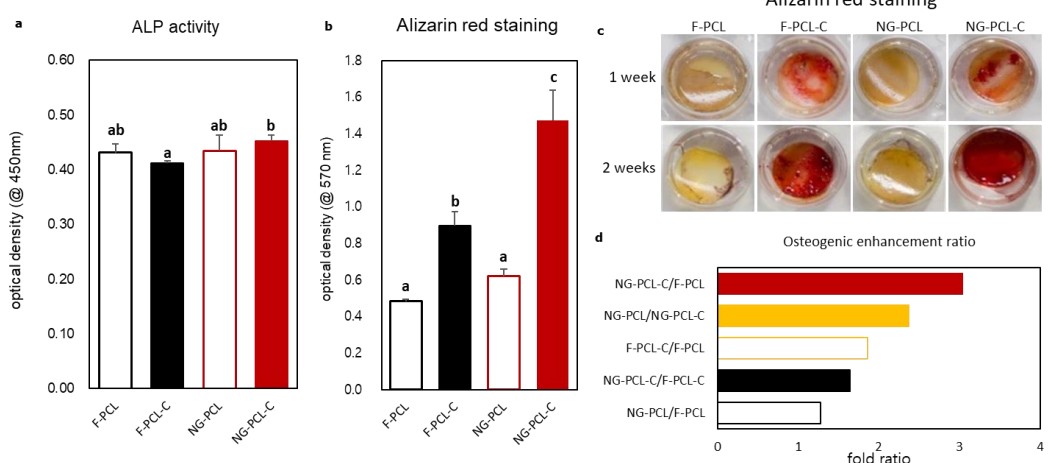

**Figure 5.** The observation of osteogenic cellular behaviors of DPSCs on the substrates. (**a**) the quantification of ALP activity (n = 5), (**b**) the quantification of mineral deposition (n = 5) (all absorbance data eliminated background signal Figure S5), (**c**) the digital camera images of mineral deposition of DPSCs on the substrates, (**d**) the relative comparisons of mineral deposition of DPSCs on the substrates. (Bars with the same letter are not significantly different according to Duncan's multiple range test at 5% probability.)

Figure 6 shows the cellular morphology of DPSCs in single-cell level on each substrate. The morphology of DPSCs was elongated on all experimental groups, and the cells on the substrate with nanopattern formed more clusters (Figure 6a). In addition, thin filopodia growing toward EBPs was observed at the end of the cell on F-PCL and F-PCL-C substrate. However, thin filopodia growth wasn't observed on NG-PCL-C. And it could be observed that the edge of DPSCs were attached along the nanopattern (Figure 6b). The elongation factors were calculated and presented in Figure 6c. It was confirmed that it had the highest value in the DPSC grown in the pattern. In Figure 6d, the cell length factors were calculated by dividing the cell length on F-PCL to each group. And it was also confirmed that the DPSCs on the NG-PCL and NG-PCL-C had a longer value than the DPSCs on the F-PCL and F-PCL-C. Figure 6e represents the ratio of the cellular area. It was found that the cell area of NG-PCL-C was significantly increased compared to other groups.

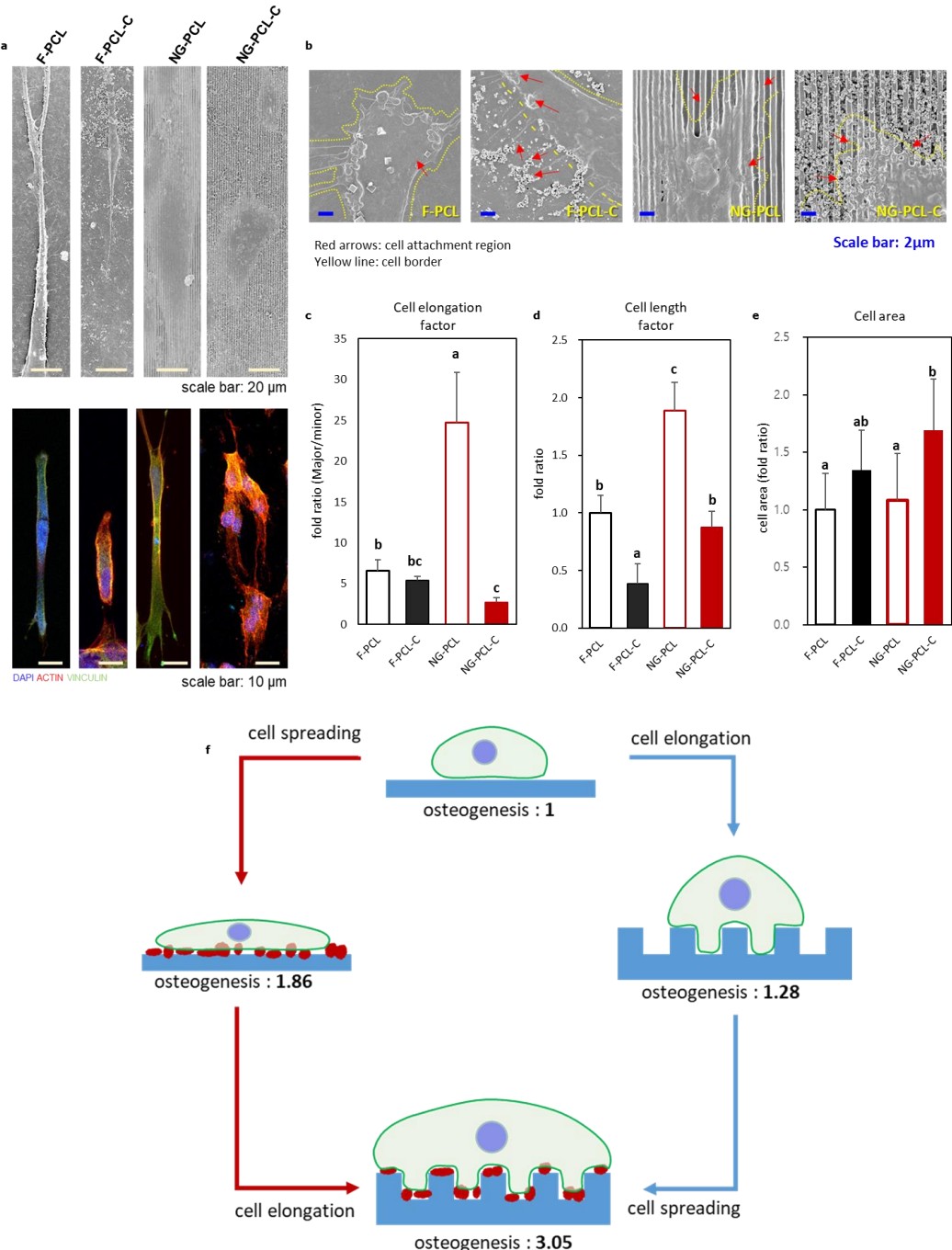

**Figure 6.** (**a**) The comparison of DPSCs morphologies on a single cell level through FE-SEM and fluorescence images. (**b**) The observation of the interaction between DPSCs and the substrates. (**c**) The calculation of elongation factors of DPSCs through the fluorescence images. (n = 10) (**d**) The relative cell length factors of DPSCs. (n = 10) (**e**) The relative cell area measures through image analysis. (n = 10) (**f**) The relative osteogenic capacities comparison between all samples. (Bars with the same letter are not significantly different according to Duncan's multiple range test at 5% probability.)

## 4. Discussion

Various implant materials for bone regeneration have been continually developed and designed to improve interaction with the patient's tissue at the cellular level through physical and chemical modification of the implant surface [36,37]. In particular, the PCL material approached in this study is a biodegradable and biocompatible polymer that is easy to fabricate in the form of a membrane and is

suitable for studying the interaction between a cell and a substrate surface. In particular, nanogrooved PCL surface with highly controlled and reproducible is a suitable platform for observing the interaction with cells. However, the surfaces coated with bioactive ceramics have been scarcely reported because bioceramics coating can hinder the spatial integrity of the substrate. Thus, the architectural role in cell interaction can be lost. In this study, we applied the bioceramics coating to a nanogrooved PCL patch with particles isolated from equine bone using a simple dip-coating method. We prepared the four different substrates upon coating and pattern which are F-PCL, F-PCL-C, NG-PCL, and NG-PCL-C. To characterize this method, surface features were studied and the effects on the proliferation and differentiation of pulp-derived stem cells were evaluated.

In this study, we designed nanogroove with 800 nm size for the fabrication of PCL patches since it has been reported that the 800 nm pattern spacing could increase cell migration, proliferation, and differentiation [38]. In addition, bone powder extracted from the equine femur bone is reported to have similar mechanical strength and osteogenesis capacity compared to other animal bones. The EBPs (particle size from 400 nm to 2 $\mu$m) were ground using a high energy ball mill. The atomic ratio of calcium increased with time of dip-coating. This result indicates that the dip-coating process is an appropriate method for the coating of bioceramics on the PCL substrates. However, the longer duration time (e.g., 10 mins) causes powder aggregation and affects the coating quality. Therefore, we conducted water contact angle measurement and observed water droplet shape to decide the degree of EBP coating. On the uncoated nanogrooved surface, the phenomenon of surface water droplets extending in the nanogroove direction was observed according to the Cassie-Baxter theory [39]. In particular, the elliptical droplets had different contact angles between the major and minor axes. The longer the shape of the ellipse, the contact angle became higher. As the coating time increased, the difference in the contact angle between the major and the minor axis decreased, and the shape of the water droplet became more circular than the uncoated group. In particular, for nanogrooved patch treated for 5 minutes with EBPs coating, we assumed that the effect of the pattern was almost eliminated because the disparity between the major and the minor axis of the water contact angle was not significantly different, and the shape of the droplet was almost circular. Based on these observations, we decided that the optimal coating time for the nanogrooved patch was 1 minute. This condition was suitable for studying the combination effect of EBPs coating and nanogroove pattern.

In general, it has been hypothesized that surfaces with nanotopography can improve the cell adhesion by increasing the surface area of the substrate because cells could recognize the morphological features of the surface such as grating, pit, hole, etc.. In our study, nanogrooves also influenced cell adhesion and morphology, as demonstrated by 12-hour cell culture [9]. In this work, DPSCs both recognized the surface topography and EBPs distributed on the substrate. Our interpretation was supported by the results of cellular behavior which include the aligned cellular morphology on nanogrooves and the cluster formed on the EBPs. Moreover, this cellular behavior also influenced the osteogenic differentiation of DPSCs. No significant difference was found in the adhesion ability of DPSCs in all groups, however, the morphology was affected by the surface characteristics of the substrate. The DPSCs cultured on the nanogroove patches were found to have a longer shape than DPSCs on F-PCL and F-PCL-C, with the elongation (major/minor axis) level of cells approximately four times higher. As cells reach confluence, MSCs become bigger and had mostly spindle-like, fibroblastic-like shape [40]. The DPSCs became fibroblastic-like shape and more aligned according to nanogroove.

In this study, the ALP activity was observed to evaluate the early stage of osteogenesis on day 7. the ALP activity results showed a significant difference between F-PCL-C and NG-PCL-C, but the difference is not much great. However, in the mineral deposition study, the significant differences in the substrate were found. Previous reports have shown that, in the case of a substrate composed of calcium phosphate (CaP) cement, CaPs in the substrate were stained in red, and it gives background color. After the cell seeding, the stained color became darker when the calcium mineral was deposited in the cells. This trend was also observed in our experiments and confirmed that higher levels of

calcium minerals were deposited at NG-PCL-C than F-PCL-C. Given that the mineral deposition of DPSCs in 2 weeks is higher than that of 1 week. We could observe that our substrate affected the promotion of DPSCs' osteogenic differentiation. According to our observation, we had developed an index that represents the degree of osteogenic differentiation of DPSCs using the quantification results of mineral deposition in 2 weeks. The effect of the nanogroove was defined as about 1.28 times, and the effect by EBP was defined as about 1.86 times.

In order to achieve successful tissue regeneration, applying a scaffold that stimulates stem cells to improve treatment efficiency is one of the ultimate goals in tissue engineering and regenerative medicine. Bioceramics, calcium phosphate-based materials such as hydroxyapatite, beta-tricalcium phosphate, and xenograft-derived bone grafts, have been used for human bone regeneration. In addition, nanogrooved surface is a platform that is constantly being explored as a new approach for effective implants. No studies have yet been reported on the behavior of human pulp-derived stem cells on nanogrooves and bioceramics materials through in vitro experiments. Here, we suggest a nanogroove/bioceramics integrated platform through the simple dip-coating method that can synergistically enhance the cell-substrate interaction. In summary, the nanogroove substrate coated with EBPs provides an adequate environment for DPSCs compared to the F-PCL group, facilitating the osteogenesis of DPSCs. This explanation was supported through the analysis of morphological features, and our insight according to these observations are briefly shown in Figure 6f.

In our platforms, EBP coating could provide a site for cells to be attached and nanogroove could induce cell elongation. This novel combined effect of each cue may enhance cell differentiation.Since endothelin and vitamin have been known as an important role of ECM, it would be efficient strategies to improve our PCL/EBP patches to alleviate the inflammatory response [30]. Our current approach could be used as an effective strategy for enhancing stem cell behaviors for clinical use such as a patch for a guided bone generation.

## 5. Conclusions

In this study, we proposed the simple mothed for the fabrication of the GBR/GTR membrane combined the nanotopographic surface and EBPs coating to enhance the osteogenic behavior of DPSCs. We confirmed that either factor alone could promote the osteogenesis of DPSCs respectively. The osteogenic phenotypes and calcium mineral deposition were further enhanced when both factors were combined. We observed that EBPs coating enhanced the surface hydrophilicity and osteogenic phenotype of DPSCs. This simple method to achieve a suitable environment for osteogenesis of dental stem cells via nanotopography and EBPs coating modulation may be regarded as a promising technique for biomaterials-based medical platforms such as dental and maxillofacial surgery applications.

**Supplementary Materials:** The following are available online at http://www.mdpi.com/2076-3417/10/7/2398/s1, Figure S1: Characterization of equine bone powders, Figure S2: Representative FE-SEM images of equine bone powders coated nanopatterned patches with coating conditions, Figure S3: Evaluation of coating conditions, Figure S4: High resolution surface image of NG-PCL with or without equine bone powder coating, and Figure S5: (a) The quantification data of EBP coating density of F-PCL-C and NG-PCL-C. The flat surface and nanogroove pattern had same amount of EBP on their surfaces. (b) The graph indicated that the ARS quantification assay of day 3 MC3T3 cell culture.

**Author Contributions:** Conceptualization, K.-J.J. and J.K.; methodology, K.-J.J., S.K., S.P. (Sangbae Park); experiment assistance, S.K., W.K., Y.G., S.P. (Sunho Park); supervison and project administration, K.-T.L., J.K. and H.S. All authors have read and agreed to the published version of the manuscript.

**Funding:** This work was supported by National Research Foundation (NRF) grants funded by the Korea Government (2019R1I1A3A0106345, 2016M3A9B4919374). This work was also supported by the Korea Institute of Planning and Evaluation for Technology in Food, Agriculture and Forestry (IPET) through the Agriculture, Food and Rural Affairs Research Center Support Program, funded by the Ministry of Agriculture, Food and Rural Affairs (MAFRA) (Project No. 714002).

**Acknowledgments:** 

**Conflicts of Interest:** The authors declare no conflict of interest.

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
