# Peer review of "Lithographically-Fabricated HA-Incorporated PCL Nanopatterned Patch for Tissue Engineering"

_applsci, doi:10.3390/app10072398_

Round 1

Reviewer 1 Report

This article describes the preparation of polycarolactone nanopatterned patches with equine bone powders. These materials have been analyzed and their behaviour in osteogenesis of human dental pulp stem cells presented. The study is very thorough examining many different aspects. The results are very promising.

The English language used is correct although there are several errors along with a few format and typographical mistakes. I indicate a few in my report but I urge the authors to recheck the manuscript as surely I have missed things.

There are a lot of abbreviations (some of which are not defined). Maybe these should be listed at the beginning or end of the paper.

In the introduction, between lines 49 and 66, references need to be given.

Correct “behaviors” to “behavior” throughout the text

Use US spelling behavior and not UK behaviour

Line 167, “we cultured the DPSCs cultured on the nPCL nanopatterned matrix with EBPs for 24 h and 7 days”. Try not to use cultured twice in the same sentence

Lines 170-173, the sentence has been repeated.

Line 183 184, correct to “the initial osteogenic differentiation marker, that there were no significant differences between all the experimental groups.” Same for line 264

Line 219, correct to “In order to overcome such drawbacks, various approaches were introduced”

Line 220 221, correct to “However, there still exists other drawbacks”

Line 309-311, “In the comparison of the same experimental group, the ratio of the length of the cell to the length of the cell is not more than twofold indicating that the minor axis of the cell is shorter in the nPCL patch.” This sentence needs to be corrected so as not to put length of the cell twice.

Line 311 312, correct to “What is special is that the cell's elongation factor was observed be the lowest as a result of morphological analysis”

Author Response

Reviewer #1

This article describes the preparation of polycarolactone nanopatterned patches with equine bone powders. These materials have been analyzed and their behaviour in osteogenesis of human dental pulp stem cells presented. The study is very thorough examining many different aspects. The results are very promising.

 The English language used is correct although there are several errors along with a few format and typographical mistakes. I indicate a few in my report but I urge the authors to recheck the manuscript as surely I have missed things.

There are a lot of abbreviations (some of which are not defined). Maybe these should be listed at the beginning or end of the paper.

[Revision]

In the introduction, between lines 49 and 66, references need to be given.

Correct “behaviors” to “behavior” throughout the text

Use US spelling behavior and not UK behaviour

Line 167, “we cultured the DPSCs cultured on the nPCL nanopatterned matrix with EBPs for 24 h and 7 days”. Try not to use cultured twice in the same sentence

Lines 170-173, the sentence has been repeated.

Line 183 184, correct to “the initial osteogenic differentiation marker, that there were no significant differences between all the experimental groups.” Same for line 264

Line 219, correct to “In order to overcome such drawbacks, various approaches were introduced”

Line 220 221, correct to “However, there still exists other drawbacks”

Line 309-311, “In the comparison of the same experimental group, the ratio of the length of the cell to the length of the cell is not more than twofold indicating that the minor axis of the cell is shorter in the nPCL patch.” This sentence needs to be corrected so as not to put length of the cell twice.

Line 311 312, correct to “What is special is that the cell's elongation factor was observed be the lowest as a result of morphological analysis”

[Response]

: We appreciate your kind comments. We revised the manuscript totally. It is also noted that the revised manuscript has been checked by a native English speaker.

Reviewer 2 Report

The authors created PCL platforms with nanogrooves and these platforms were coated with equine bone powder, through dipping, to facilitate dental pulp stem cells (DPSC) into bone-like cells through osteogenesis. The claim was that the synergistic effects of the nano topology and the surface coating increased DPSC differentiation into bone-like cells.

Overall, this is an important topic as it may well aid tissue regeneration in dental applications. The manuscript in the present form requires much work. The language needs to be checked throughout, and it is riddled with inconsistencies, to highlight a few:

  • e.g.1 Pg 3 : It is said a PUA mother mold was prepared, and subsequently there is mention of PDMS mold.
  • e.g.2 Pg 3 : cell viability was assessed at 6 h and 7 days, but there are not results for 6 h cell viability.
  • What is nPCL - nanogroove PCL? please define at first use.

The following points need to be addressed:

Nanogrooves:

  • The limited background mentioned in the introduction did suggest some sizes for nanogrooves. The authors however did not mention, or experimentally ascertain, why they used nanogroove size of 800 nm. 
  • The authors did observe this, air pockets, during water contact angle. Because of the dimension of these nanogrooves, capillary action might cause these artefacts and affect the results. The authors should place these membranes (all conditions) in a desiccator and submerge them with cell culture media (+FBS) and dried. Water contact angle should then be tested after protein absorption to determine the actual effect of the bone powder vs PCL.
  • From the SEM images, the bone powder are large in size (figure 1b). The claim on page 9 " surface chemistry vs surface roughness is a bit far fetched. The large sized bone particle contributes to surface roughness as well, unless the ball milling produces very fine particles that actually gives a uniform coating, you cannot exclude that the bone powder contributes to roughness. The claim of nano-size should also be validated, since the ball milling will give a range of sizes.
  • Perhaps this paper by the Chen group can give clues about the particle spacing and osteogenesis.

DPSC:

  • The WST-1 results were normalized against Flat PCL? this has to be made clear.
  • Could the authors give a general sense of how many cells attach on their biomaterials, as compared to tissue culture polystyrene, as these surfaces are optimized for cell attachment. it will not be a good material if 10X the number of cells are required when compared to tissue culture polystyrene.
  • Is the osteogenic enhancement ratio in Figure 5 = osteogenesis number in Figure 6? Please detail all of these calculation methods in Methods.

Presentation:

  • Figure1. The fonts are small and there is no written information to facilitate understanding of the fabrication process. What is the last image after fabrication showing (a) what do the colors there represent? 
  • there is no form of statistical analyses throughout. This should be performed for all comparison and detailed in Methods.
  • All the figure caption should contain details that allow the reader to comprehend the figure w/o the main text.
  • Discussion should be re-worked, it seems to be repeating itself incessantly. Huge portion of discussion should be in introduction to impact the need for such a study.

Author Response

Reviewer #2

The authors created PCL platforms with nanogrooves and these platforms were coated with equine bone powder, through dipping, to facilitate dental pulp stem cells (DPSC) into bone-like cells through osteogenesis. The claim was that the synergistic effects of the nano topology and the surface coating increased DPSC differentiation into bone-like cells.

Overall, this is an important topic as it may well aid tissue regeneration in dental applications. The manuscript in the present form requires much work. The language needs to be checked throughout, and it is riddled with inconsistencies, to highlight a few:

[Revision]

e.g.1 Pg 3: It is said a PUA mother mold was prepared, and subsequently there is mention of PDMS mold.

e.g.2 Pg 3: cell viability was assessed at 6 h and 7 days, but there are not results for 6 h cell viability.

What is nPCL - nanogroove PCL? please define at first use.

[Response] : We appreciate your kind comments. We have revised the manuscript as follows: “Then, the assembly comprising the PCL layer and PDMS mold was cooled at 25 °C for 30 min, and the PDMS mold was peeled off from the PCL, and named NG-PCL. In addition, flat PCL patches were prepared using the flat PDMS mold through the same procedure and named F-PCL.”

The following points need to be addressed:

  1. Nanogrooves:

The limited background mentioned in the introduction did suggest some sizes for nanogrooves. The authors however did not mention, or experimentally ascertain, why they used nanogroove size of 800 nm.

[Response]

: We appreciate your kind comments. We added the reason why we selected the 800 nm nanogroove substrates in this work.

[Revision]

“In this study, we designed nanogroove with 800 nm size for fabrication of PCL patches since it has been reported that the 800 nm pattern spacing could increase cell migration, proliferation, and differentiation [19].”

  1. Kim, D.-H., E.A. Lipke, P. Kim, R. Cheong, S. Thompson, M. Delannoy, K.-Y. Suh, L. Tung and A. Levchenko, 2010. Nanoscale cues regulate the structure and function of macroscopic cardiac tissue constructs. Proceedings of the National Academy of Sciences, 107(2): 565-570
  2. The authors did observe this, air pockets, during water contact angle. Because of the dimension of these nanogrooves, capillary action might cause these artefacts and affect the results. The authors should place these membranes (all conditions) in a desiccator and submerge them with cell culture media (+FBS) and dried. Water contact angle should then be tested after protein absorption to determine the actual effect of the bone powder vs PCL.

[Response]

The water contact angle after soaking in FBS incorporated culture medium had 0 degree in all experimental groups with or without EBPs. The water contact angle after soaking in FBS incorporated culture medium had 0° in all experimental groups with or without EBPs. It seems that the protein coating enhances the hydrophilicity of all specimens and could remove the air pocket effects. Therefore, the equine bone power coating effects could be interpreted as enhancing osteogenic effects without improving hydrophilicity.

  1. From the SEM images, the bone powder is large in size (figure 1b). The claim on page 9 " surface chemistry vs surface roughness is a bit far fetched. The large sized bone particle contributes to surface roughness as well, unless the ball milling produces very fine particles that actually gives a uniform coating, you cannot exclude that the bone powder contributes to roughness. The claim of nano-size should also be validated, since the ball milling will give a range of sizes.

[Response]

: We appreciate your very insightful comments. We have added the discussion about the issue of surface chemistry and roughness. Particle sizes were measured by using particle size distributer, and the results were attached to the supplementary figure.1. The diameter on cumulative (d0.1, d0.5, d0.9) of EBPs is 0.779 ± 0.099, 4.13 ± 1.71, and 19.40 ± 0.2.47 μm, respectively.

And we added this sentences at line 169-171:

 “The detailed characteristics of EBPs showed supplementary figure 1 and The diameter on cumulative (d0.1, d0.5, d0.9) of EBPs is 0.779 ± 0.099, 4.13 ± 1.71, and 19.40 ± 0.2.47 μm, respectively”

Figure S.1. Characterization of equine bone powders. (a) Size distribution of equine bone powders. (b) A representative FE-SEM image of equine bone powders used in this work.

[Revision]

DPSC:

4.The WST-1 results were normalized against Flat PCL? this has to be made clear.

[Response]

: The WST-1 results were normalized against the results of F-PCL. We have added it in the ‘Materials and Methods’ part in the revised manuscript.

[Revision]

  1. Could the authors give a general sense of how many cells attach on their biomaterials, as compared to tissue culture polystyrene, as these surfaces are optimized for cell attachment. it will not be a good material if 10X the number of cells is required when compared to tissue culture polystyrene.

Is the osteogenic enhancement ratio in Figure 5 = osteogenesis number in Figure 6? Please detail all of these calculation methods in Methods.

[Response]

: We have added the method for the calculation of osteogenic enhancement ratio in the ‘Materials and Methods’ part in the revised manuscript.

Presentation:

[Revision]

Figure1. The fonts are small and there is no written information to facilitate understanding of the fabrication process. What is the last image after fabrication showing (a) what do the colors there represent? 

there is no form of statistical analyses throughout. This should be performed for all comparison and detailed in Methods.

[Response]
: We addded the detailed description of the figures in the revised manuscript. And the indications of statistical significance were re-written in the figure, and the method used for the statistics was added in the ‘Materials and Methods’ part.

All the figure caption should contain details that allow the reader to comprehend the figure w/o the main text.

Discussion should be re-worked; it seems to be repeating itself incessantly. Huge portion of discussion should be in introduction to impact the need for such a study.

[Response]

: We appreciate your kind comments. We have revised it.

Reviewer 3 Report

The manuscript describes a PCL nanopatterned patch with an EBP coating and demonstrated DPSC osteogenesis on the patch. The system is characterized well in terms of its surface hydrophobicity characteristics as well as cellular biochemical and morphological features. The results in the manuscript make sense given our current understanding of cellular behaviors; these include cell spreading aligned with the nanogroove on the patterned surfaces etc. The differences in osteogenesis, the potential synergy between the topology and coating and consequent results in Fig 5 are very insightful.

English

I would recommend extensive editing of the grammar in the manuscript. 

Major comments:

  • Could you quantify the uniformity of EBP coating on the nanopatterned vs flat surfaces? Are there differences in the coating that cause the differences in cell behavior observed? Could you rule this out with some sort of staining/imaging?
  • 155-156: could you quantify the smaller cell adhesion area? Please include in another figure, add the statistical tests required.
  • 160-161: presumably some cells did not adhere as well to the uncoated surfaces and would have died. Were these included in the cell viability test or excluded? This could affect the measurement. Please include a more specific description of this in the methods section
  • 172-173: is there a hypothesis for why orientation was better at day 7 vs day 1? Could it be cell laid matrix? Could you stain for cell laid matrix?

Minor comments:

  • Line 107: perhaps you meant 12h instead of 6h? Because Fig 3 indicates 12
  • Fig 2A: please add a scalebar to show scale
  • Fig 2B, Fig 5a/b, Fig 6C/D: please add stats, p-values
  • 200-204: please add definitions of relative ratio and cell length factor earlier in the methods section
  • 219: how do you define "cell response". Please be more specific

Author Response

Reviewer #3

The manuscript describes a PCL nanopatterned patch with an EBP coating and demonstrated DPSC osteogenesis on the patch. The system is characterized well in terms of its surface hydrophobicity characteristics as well as cellular biochemical and morphological features. The results in the manuscript make sense given our current understanding of cellular behaviors; these include cell spreading aligned with the nanogroove on the patterned surfaces etc. The differences in osteogenesis, the potential synergy between the topology and coating and consequent results in Fig 5 are very insightful.

 English

I would recommend extensive editing of the grammar in the manuscript. 

: Thank you for your comment. It is noted that the revised manuscript has been checked by a native English speaker.

 Major comments:

[Revision]

  1. Could you quantify the uniformity of EBP coating on the nanopatterned vs flat surfaces? Are there differences in the coating that cause the differences in cell behavior observed? Could you rule this out with some sort of staining/imaging?

[Response]

: Thank you so much for your kind comments. We have added the results of coating uniformity nanopatterned and flat surface through alizarin staining methods. The Alizarin Red S (ARS) quantification was performed by ARS extraction methods. The results showed in the supplementary Figure 5(a).

Figure S.5. (a) The quantification of EBP coating density of F-PCL-C and NG-PCL-C. The flat surface and nanogroove pattern had same amount of EBP on their surfaces. (b) ARS quantification assay of day 3 MC3T3 cell culture (cell confluent on patches). All ARS quantification results had subtracted this value. (n = 5).

[Revision]

  1. 155-156: could you quantify the smaller cell adhesion area? Please include in another figure, add the statistical tests required.

[Response]

: The Multi Cell Outliner that is ImageJ plugin was used to quantify the cell area. The cellular edge was detected by this software and calculated the cellular area using the number of pixels in the image. The results were added in the Figure 6e.

Figure 6. (a) The comparison of DPSCs morphologies on a single cell level through FE-SEM and fluorescence images. (b) The observation of the interaction between DPSCs and the substrates. (c) The calculation of elongation factors of DPSCs through the fluorescence images. (n=10) (d) The relative cell length factors of DPSCs. (n=10) (e) The relative cell area measures through image analyze. (n=10) (f) The relative osteogenic capacities comparison between all samples.

[Revision]

  1. 160-161: presumably some cells did not adhere as well to the uncoated surfaces and would have died. Were these included in the cell viability test or excluded? This could affect the measurement. Please include a more specific description of this in the methods section.

[Response]

: The WST-1 assay was performed after washing the samples twice with PBS. We have added it in the materials and methods section.

[Revision]

  1. 172-173: is there a hypothesis for why orientation was better at day 7 vs day 1? Could it be cell laid matrix? Could you stain for cell laid matrix?

[Response]

: It is probably due to decreased cell-to-cell spacing. We have added it in the discussion with some references.

[Revision]

Line 107: perhaps you meant 12h instead of 6h? Because Fig 3 indicates 12

Fig 2A: please add a scalebar to show scale

Fig 2B, Fig 5a/b, Fig 6C/D: please add stats, p-values

200-204: please add definitions of relative ratio and cell length factor earlier in the methods section

219: how do you define "cell response". Please be more specific

[Response]

: We appreciate your very kind comment. We have revised them.

Reviewer 4 Report

The paper titled 'Lithographically fabricated HA-incorporated PCL nanopatterned patch for tissue engineering' by Jang et al highlights the combinatorial approach of fabricating EBP-coated nanopatterned PCL membranes. The research article highlights the importance of surface physical and chemical modification of otherwise hydrophobic PCL membranes for improved osteogenic differentiation of DPSCs.

The article provides description of membrane fabrication and cell culture, however they fail to provide detailed characterization of EBP powder collected after thermal treatment and milling. 

The authors also need to provide detailed description of how the coating time of EBP alters the surface properties of coated surface. They mention that they coated it for different times but there is no mention of what they chose or how different coating times altered surface physical properties and thereby biological response.

One more factor that needs to be considered is the effect of surface roughness. There is no mention in the article about how this parameter is affected during surface patterning and coating.

High resolution SEM micrographs need to be provided as it is hard to see surface details esp. after EBP coating.

The authors don't mention anything about osteogenic differentiation in the methodology section and then they have a whole section in the results. 

The authors comment that there is no difference in ALP, which is an early marker for osteogenic differentiation, what time point was the measurement done? What is the ALP expression at an earlier time point, like day 7 or so.

The authors see a lot of alizarin red staining for alizarin red. They also mention that there can be background staining for CaP but don't provide any evidence that the stain observed is due to the mineralized matrix secreted by the cells. What about other osteogenic markers, like osteocalcin, osteopontin etc. 

Author Response

Reviewer #4

The paper titled 'Lithographically fabricated HA-incorporated PCL nanopatterned patch for tissue engineering' by Jang et al highlights the combinatorial approach of fabricating EBP-coated nanopatterned PCL membranes. The research article highlights the importance of surface physical and chemical modification of otherwise hydrophobic PCL membranes for improved osteogenic differentiation of DPSCs.

The article provides description of membrane fabrication and cell culture; however they fail to provide detailed characterization of EBP powder collected after thermal treatment and milling. 

[Revision]

The authors also need to provide detailed description of how the coating time of EBP alters the surface properties of coated surface. They mention that they coated it for different times but there is no mention of what they chose or how different coating times altered surface physical properties and thereby biological response.

One more factor that needs to be considered is the effect of surface roughness. There is no mention in the article about how this parameter is affected during surface patterning and coating.

[Response]

: We appreciate your comment. We added the information about EBPs with supplementary figures in the revised manuscript.

Figure S1. Characterization of equine bone powder. (a) The graph indicates particle size distribution of equine bone specimen. The diameter on cumulative (d0.1, d0.5, and d0.9) of EBPs is 0.779 ± 0.099, 4.13 ± 1.71, and 19.40 ± 0.2.47 μm, respectively (n = 5) (b) The representative FE-SEM observation of equine bone powders.

[Revision]

High resolution SEM micrographs need to be provided as it is hard to see surface details esp. after EBP coating.

[Response]

: We added the high-resolution surface image about nanogrooved and EBP coated nanogroove as supplementary figure 4.

Figure S.4. High resoloution surface image of NG-PCL with or without equine bone powder coating

[Revision]

The authors don't mention anything about osteogenic differentiation in the methodology section and then they have a whole section in the results.

[Response]

: We added the methodology of evaluation of osteogenic differentiation in materials and methods.

[Revision]

The authors comment that there is no difference in ALP, which is an early marker for osteogenic differentiation, what time point was the measurement done? What is the ALP expression at an earlier time point, like day 7 or so.

[Response]

: ALP concentration was measured on the 7-day cell culture with osteogenic media.

[Revision]

The authors see a lot of alizarin red staining for alizarin red. They also mention that there can be background staining for CaP but don't provide any evidence that the stain observed is due to the mineralized matrix secreted by the cells. What about other osteogenic markers, like osteocalcin, osteopontin etc. 

[Response]

: We added the stained patches' images and destaining results in the figure. And the background absorbances were eliminated from the ARS experiments for 2 weeks.

Figure S.5. (a) The quantification data of EBP coating density of F-PCL-C and NG-PCL-C. The flat surface and nanogroove pattern had same amount of EBP on their surfaces. (b) The graph indicated that the ARS quantification assay of day 3 MC3T3 cell culture (cell confluent on patches). All ARS quantification results had subtracted this value. (n = 5).

Round 2

Reviewer 2 Report

changes are to my satisfaction.

Author Response

Reviewer #2

changes are to my satisfaction.

[Response]

We appreciate your comment. We revised our manuscript

Reviewer 3 Report

Thank you for your revision. All your responses make sense to me.

Please do another minor spell check of the manuscript.

Author Response

Reviewer #3

Thank you for your revision. All your responses make sense to me.

Please do another minor spell check of the manuscript.

[Response]

We appreciate your comment. We revised our manuscript
